# Domino Nitro Reduction-Friedländer Heterocyclization for the Preparation of Quinolines

**DOI:** 10.3390/molecules27134123

**Published:** 2022-06-27

**Authors:** Kwabena Fobi, Richard A. Bunce

**Affiliations:** Department of Chemistry, Oklahoma State University, Stillwater, OK 74078-3071, USA; kfobi@okstate.edu

**Keywords:** domino reaction, Friedländer synthesis, dissolving metal reduction, heterocyclization, quinolines, quinolin-2(1*H*)-ones

## Abstract

The Friedländer synthesis offers efficient access to substituted quinolines from 2-aminobenzaldehydes and activated ketones in the presence of a base. The disadvantage of this procedure lies in the fact that relatively few 2-aminobenzaldehyde derivatives are readily available. To overcome this problem, we report a modification of this process involving the in situ reduction of 2-nitrobenzaldehydes with Fe/AcOH in the presence of active methylene compounds (AMCs) to produce substituted quinolines in high yields. The conditions are mild enough to tolerate a wide range of functionality in both reacting partners and promote reactions not only with phenyl and benzyl ketones, but also with β-keto-esters, β-keto-nitriles, β-keto-sulfones and β-diketones. The reaction of 2-nitroaromatic ketones with unsymmetrical AMCs is less reliable, giving a competitive formation of substituted quinolin-2(1*H*)-ones from the cyclization of the *Z* Knoevenagel intermediate which appears to be favored when certain large groups are adjacent to the AMC ketone carbonyl.

## 1. Introduction

The Friedländer synthesis provides an efficient synthetic approach to 3-substituted quinoline derivatives from 2-aminobenzaldehydes and activated ketones [1]. A number of modified variants of the original reaction have also been reported to access the quinoline system [2,3]. However, a disadvantage of the classic reaction is the relatively limited number of commercially available 2-aminobenzaldehydes [4,5]. Two alternative methods have been developed in an effort to overcome this limitation involving the in situ oxidation of 2-aminobenzyl alcohols to the requisite 2-aminoaldehydes using ruthenium-based oxidants [6,7]. We wish to report our use of iron (Fe) in acetic acid (AcOH) to reduce 2-nitrobenzaldehydes and ketones to the 2-amino derivatives and heterocyclization with β-keto-esters, β-keto-nitriles, β-keto-sulfones, benzyl ketones, phenyl ketones and β-diketones in a one pot procedure. A search of commercial sources indicated that there are significantly more 2-nitrobenzaldehyde derivatives available than the more reactive 2-aminobenzaldehydes. Several previous reports documented the reduction of 2-nitrobenzaldehydes to 2-aminoaldehydes prior to cyclization, some of which are described in the following: one utilized a Hantzsch 1,4-dihydropyridine as the source of hydrogen [8] and two others utilized Fe/AcOH with substituted benzaldehydes [9,10]. The current study includes a wider range of functional groups in the activated ketones than the earlier work and further extends the reaction to 2-nitroaromatic ketones.

Dissolving metal reduction with Fe/AcOH is a very mild and selective method for the reduction of nitroaromatics to anilines. As such, it has been used as an initiating reaction for several other domino reaction schemes, including a synthesis of indoles [11], a synthesis of tetrahydroquinolines [12], a synthesis of dihydrodibenzazepinones [13] and in several syntheses of carbazolones [14,15,16]. In these reactions, the nitroarene is reduced in the presence of a carbonyl compound or a Michael acceptor, and the resulting amino group reacts with the ketone or activated double bond to close a ring. 

The Friedländer synthesis involves two sequential reactions. Initially, a benzylic ketone undergoes Knoevenagel condensation with the aldehyde of 2-aminobenzaldehyde to produce the conjugated product. This is followed by the addition of the amino function to the ketone carbonyl and loss of water to aromatize the newly formed ring. Due to free rotation in the initial carbonyl addition product and reversibility of the condensation, the proper double bond geometry can be achieved to position the carbonyl for ring closure with the aniline nitrogen. An alternative process via the Schiff base derived from the ketone and the aniline is also possible. Classically, Friedländer heterocyclization has been performed under basic conditions (HO^−^ or RO^−^) [5], but acidic conditions (HCl and MeSO_3_H) have been reported [17,18,19], and in one case, no added catalyst was required [20]. 

In the present work, the Friedländer sequence was initiated by dissolving metal reduction of the nitroaromatic carbonyl compounds with Fe/AcOH in the presence of β-keto-esters, β-keto-nitriles, β-keto-sulfones, benzyl ketones and β-diketones. Without Fe to provide a source of electrons and a proton source, the reaction does not proceed from the nitro compounds. In the presence of Fe/AcOH, however, in situ reduction of the nitro occurs and is followed by heterocyclization. The use of catalytic hydrogenation in this procedure is not possible due to the competitive reduction of the double bond in the Knoevenagel intermediate, but this double bond is stable to the Fe/AcOH. In the course of this project, a representative Friedländer synthesis using 2-aminobenzaldehyde and ethyl acetoacetate was performed in AcOH, and the reaction proceeded to form quinolines in a nearly quantitative yield. Thus, AcOH is an excellent solvent for this transformation. 

Quinolines express a multitude of biological activities and are valuable in the treatment of malaria [21,22,23,24,25,26,27] and other tropical diseases (e.g., Chagas disease, human African trypanosomiasis and leishmaniasis) [28] as well as tuberculosis [29], cancer [30,31] and bacterial infections [32]. Several quinoline-based drugs have been known for many years, while others are more experimental in nature. Several examples of drugs incorporating quinoline as the core ring structure are depicted in Figure 1 below.

## 2. Results and Discussion

The 2-nitrobenzaldehyde (**1a**), 5-fluoro-2-nitrobenzaldehyde (**1b**), 5-methoxy-2-nitrobenzaldehyde (**1c**) and 2-nitroacetophenone (**1d**) substrates were commercially available. However, 2-nitrobenzophenone (**1e**), while commercial, was prohibitively expensive, and thus, was prepared according to the literature method [33,34]. An outline for the representative reaction of 2-nitrobenzaldehyde (**1a**) with 2,4-pentanedione (**2**), a symmetrical AMC, is shown in Figure 1. The reaction was performed using 1 equiv. of 2-nitrobenzaldehyde (**1a**) and 3 equiv. of the 2,4-pentanedione (**2**) in glacial AcOH. The reactants were heated to 95–110 °C, and 4 equiv. (relative to the **1a**) of Fe powder (<100 mesh) were added. The mixture turned brown, and a tan precipitate was noted. The reaction was heated for 3–4 h, at which time thin-layer chromatography indicated the complete disappearance of **1a** with full conversion to the 2-aminobenzaldehyde (**3a**) and ring closure of the Knoevenagel intermediate **4**. Work-up and column chromatography afforded the pure heterocycle **5** in high yields. 

The mild reduction conditions were found to be highly tolerant of other functionalities in the substrates. In addition to reacting with benzyl and phenyl ketones, **1a** was reduced to **3a** in the presence of β-keto-esters, β-keto-nitriles, β-keto-sulfones and β-diketones. All of these substrates underwent clean reactions to deliver the substituted quinolines (see Table 1). Although chromatography was required to isolate analytically pure material, the reaction using **1a** afforded only the expected quinolines with both symmetrical and unsymmetrical AMC derivatives. Since the initial condensation event between the aldehyde and a β-keto-ester or β-keto-nitrile (unsymmetrical AMCs) could yield *E* and *Z* Knoevenagel products, two heterocycles might have been expected from these substrates. However, only the quinoline product was produced, and none of the possible lactam via the cyclization of the amino group with the ester or nitrile was observed. Thus, an equilibration is required to assure that the quinoline is formed. This could occur by (1) a reverse Knoevenagel followed by recombination or possibly (2) protonation of the double bond followed by bond rotation and loss of the proton. Furthermore, no hydrolysis was observed when an ester or nitrile was present in the AMC, and no loss of the sulfonyl group occurred when this activating group was part of the AMC. With unsymmetrical β-diketones where two modes of cyclization were possible, the final cyclization occurred with the least hindered carbonyl and no self-condensation of these substrates was observed. Finally, electron withdrawing (F) and electron donating (OCH_3_) substitutions were permitted on the aromatic ring of the 2-nitrobenzaldehyde reactant (see Table 2 and Table 3). These observations confirmed that (1) the dissolving metal conditions used for the current process are very mild and selective for the nitro function, and (2) the AcOH medium permits the facile equilibration of the intermediate Knoevenagel adduct to favor the exclusive formation of the quinoline.

The mechanism of the reaction occurs in the following three stages: (1) reduction of the nitro; (2) Knoevenagel condensation of the active methylene compound with the 2-aminoaromatic carbonyl to give a 2-aminocinnamyl intermediate; and (3) ring closure of the aniline nitrogen with the ketone of the active methylene compound. The dissolving metal reduction involves sequential electron transfer to give a radical anion followed by protonation, a sequence which is repeated until the nitro is reduced to the amine. The overall conversion requires six electrons and six protons for each nitro group. Details of the dissolving metal reduction of the nitroaromatic to the aniline are outlined in Figure 2 [31]. The mechanism for the acid catalyzed Knoevenagel and ring closure with the symmetrical diketone 2,4-pentanedione (**2**) is summarized in Figure 3. Following the reduction of **1a**, the aminoaldehyde **3a** undergoes Knoevenagel condensation with **2** to give **4** via intermediate **A**. While product **4** is symmetrical, the equilibration of adducts from unsymmetrical AMCs may be necessary to bring the ketone carbonyl cis to the aminophenyl group. Subsequent condensation of the amino nitrogen with the protonated ketone of **4** (intermediate **B**) would then give aminoalcohol **C**, which would protonate to produce **D** and lose water to aromatize the quinoline ring in **5**.

The Friedländer annulation using nitroaromatic ketones **1d** and **1e** was found to be less reliable in providing the desired heterocycles, though several highly substituted quinolines were prepared (see Table 4 and Table 5, respectively). While benzyl ketones, phenyl ketones, β-diketones and a few of the unsymmetrical AMCs successfully yielded quinolines, complications arose during the cyclization of the Knoevenagel products from several β-keto-esters and nitriles. Figure 4 outlines this process for 2-nitrobenzophenone (**1e**) with ethyl benzoylacetate and benzoylacetonitrile. Following the reduction of the nitro group, these unsymmetrical substrates gave Knoevenagel condensation intermediates **E** and **F** favoring the isomer having the ester or nitrile cis to the 2-aminoaromatic ring. These intermediates did not readily equilibrate but rather cyclized to produce the substituted quinolin-2(1*H*)-one product **6** (Table 4, entries 1 and 2 and Table 5, entries 2 and 3). This outcome was evidenced by the appearance of a weak N-H absorption and a highly conjugated amide carbonyl in the FT-IR. The ^1^H NMRs also showed an N-H signal at δ 11–13 and the ester alkoxy was lost from the β-keto-ester. The ^13^C NMRs further revealed amide and ketone carbonyls at δ 160–165 and δ 198–207, respectively. Attempts to use β-keto-sulfones as AMCs failed to produce the desired quinolines in reactions with **1d** or **1e**. From these precursors, several products were formed, but none were identified.

## 3. Experimental Section

### 3.1. General Methods

Unless otherwise indicated, all reactions were performed under dry N_2_ in oven-dried glassware. All reagents and solvents were used as received. 2-Nitrobenzophenone (**1e**) was prepared using the literature procedure [33,34]. Reactions were monitored by thin layer chromatography on Analtech No 21,521 silica gel GF plates (Newark, DE, USA). Preparative separations were performed by flash chromatography on silica gel (Davisil^®^, grade 62, 60–200 mesh) containing 0.5% of UV-05 UV-active phosphor (both from Sorbent Technologies, Norcross, GA, USA) slurry packed into quartz columns. Band elution for all chromatographic separations was monitored using a hand-held UV lamp (Fisher Scientific, Pittsburgh, PA, USA). Wash solutions used in work-up procedures were all aqueous. Melting points were obtained using a MEL-TEMP apparatus (Cambridge, MA, USA) and are uncorrected. FT-IR spectra were run as thin films on NaCl disks using a Nicolet iS50 spectrophotometer (Madison WI, USA). ^1^H- and ^13^C-NMR spectra were measured using a Bruker Avance 400 system (Billerica, MA, USA) at 400 MHz and 101 MHz, respectively, in the indicated solvents containing 0.05% (CH_3_)_4_Si as the internal standard; the coupling constants (*J*) are given in Hz. Low-resolution mass spectra were obtained using a Hewlett-Packard Model 1800A GCD GC-MS system (Palo Alto, CA, USA). Elemental analyses (±0.4%) were determined by Atlantic Microlabs (Norcross, GA, USA) and are provided only for new compounds (see Appendix A).

### 3.2. General Procedure for Domino Reduction-Heterocyclization

To a solution of the 2-nitrobenzaldehyde (1.32 mmol, 1 equiv.) in AcOH (10 mL) under N_2_ the active methylene compound was added as follows: 3 equiv. for all active ketones except ethyl benzoylacetate, deoxybenzoin, 1-benzoylacetone, dimedone and methyl 4-phenylacetoacetate where 2 equiv. were used. The mixture was stirred for 15 min at 95–110 °C before the addition of Fe (<100 mesh, 4 equiv. relative to the nitro compound). When TLC (20% EtOAc in hexane) indicated the complete consumption of the starting material (3–6 h), unreacted Fe was removed by filtration through Celite before the solution was diluted with ether (50 mL) and washed with water (3 × 30 mL) to remove the AcOH. The ether was washed with NaHCO_3_ (2 × 25 mL) and saturated NaCl (1 × 25 mL), and then dried (Na_2_SO_4_). Removal of the solvent under vacuum gave a crude product, which was further purified by column chromatography (25 cm × 2 cm) using increasing concentrations of ethyl acetate (5–15%) in hexanes to afford analytical samples of the heterocyclic products. The compounds prepared are given in Table 1, Table 2, Table 3, Table 4 and Table 5 and are identified by experiment number.

### 3.3. Reactions with 2-Nitrobenzaldehyde (***1a***)

#### 3.3.1. Ethyl 2-Methylquinoline-3-carboxylate

Yield: 0.70 g (98%) as a white solid, m.p. 68–70 °C (lit. [36] m.p. 67–68 °C); IR: 1708 cm^−1^; ^1^H NMR (400 MHz, CDCl_3_): δ 8.74 (s, 1H), 8.04 (d, *J* = 8.5 Hz, 1H), 7.87 (d, *J* = 8.1 Hz, 1H), 7.78 (t, *J* = 7.7 Hz, 1H), 7.54 (t, *J* = 7.5 Hz, 1H), 3.00 (s, 3H), 4.45 (q, *J* = 7.1 Hz, 2H), 1.45 (t, *J* = 7.1 Hz, 3H); ^13^C NMR (101 MHz, CDCl_3_): δ 166.6, 158.5, 148.6, 139.9, 131.7, 128.6, 128.5, 126.5, 125.8, 124.0, 61.4, 25.7, 14.3; MS (*m*/*z*): 215 (M^+^).

#### 3.3.2. Ethyl 2-(Trifluoromethyl)quinoline-3-carboxylate

Yield: 0.32 g (90%) as a light yellow solid, m.p. 67–69 °C; IR: 1730, 1134, 1105 cm^−1^; ^1^H NMR (400 MHz, CDCl_3_): δ 8.70 (s, 1H), 8.26 (d, *J* = 7.6 Hz, 1H), 7.97 (d, *J* = 8.2 Hz, 1H), 7.91(td, *J* = 8.2, 1.4 Hz, 1H), 7.75 (t, *J* = 7.6, 1.2 Hz, 1H), 4.48 (q, *J* = 7.1 Hz, 2H), 1.44 (t, *J* = 7.1 Hz, 3H); ^13^C NMR (101 MHz, CDCl_3_): δ 165.6, 146.9, 144.7 (q, *J* = 35.2 Hz), 140.1, 132.4, 130.1, 129.6, 128.2, 127.5, 124.1, 121.1 (q, *J* = 275.6 Hz), 62.5, 14.0; MS (*m*/*z*): 269; Anal. Calcd for C_13_H_10_F_3_NO_2_: C, 58.00; H, 3.74; N, 5.20. Found: C, 57.79; H, 3.71; N, 5.07.

#### 3.3.3. Methyl 2-Ethylquinoline-3-carboxylate

Yield: 0.25 g (87%) as a light yellow oil; IR: 1733 cm^−1^; ^1^H NMR (400 MHz, CDCl_3_): δ 8.69 (s, 1H), 8.06 (d, *J* = 8.4 Hz, 1H), 7.84 (d, *J* = 8.2 Hz, 1H), 7.77 (t, *J* = 7.7 Hz, 1H), 7.52 (t, *J* = 7.5 Hz, 1H), 3.98 (s, 3H), 3.35 (q, *J* = 7.1 Hz, 2H), 1.39 (q, *J* = 7.1 Hz, 3H); ^13^C NMR (101 MHz, CDCl_3_): δ 167.1, 163.1, 148.8, 140.1, 131.6, 128.7, 128.4, 126.5, 125.6, 123.3, 52.4, 31.0, 14.0; MS (*m*/*z*): 215; Anal. Calcd for C_13_H_13_NO_2_: C, 72.54; H, 6.09; N, 6.51. Found: C, 72.61; H, 6.06; N, 6.40.

#### 3.3.4. Methyl 2-Pentylquinoline-3-carboxylate

Yield: 0.28 g (82%) as a light yellow oil; IR: 1735 cm^−1^; ^1^H NMR (400 MHz, CDCl_3_): δ 8.70 (s, 1H), 8.06 (d, *J* = 8.5 Hz, 1H), 7.85 (d, *J* = 8.1 Hz, 1H), 7.71 (t, *J* = 7.7 Hz, 1H), 7.54 (t, *J* = 8.1 Hz, 1H), 3.98 (s, 3H), 3.32 (t, *J* = 7.9 Hz, 2H), 1.78 (quintet, *J* = 7.5 Hz, 2H), 1.44 (m, 4H), 0.91 (t, *J* = 7.1 Hz, 3H); ^13^C NMR (101 MHz, CDCl_3_): δ 167.2, 162.3, 148.7, 140.1, 131.6, 128.7, 128.4, 126.5, 125.7, 123.6, 52.4, 37.8, 32.1, 30.0, 22.6, 14.1; MS (*m*/*z*): 257 (M^+^); Anal. Calcd for C_16_H_19_NO_2_: C, 74.68; H, 7.44; N, 5.44. Found: C, 74.77; H, 7.42; N, 5.35.

#### 3.3.5. Methyl 2-Isopropylquinoline-3-carboxylate

Yield: 0.30 g (99%) as a light yellow oil; IR: 1732 cm^−1^; ^1^H NMR (400 MHz, CDCl_3_): δ 8.59 (s, 1H), 8.07 (d, *J* = 8.5 Hz, 1H), 7.81 (d, *J* = 8.1 Hz, 1H), 7.74 (t, J = 8.1 Hz, 1H), 7.51 (t, *J* = 8.1 Hz, 1H), 3.99 (septet, *J* = 6.8 Hz, 1H), 3.97 (s, 3H), 1.40 (d, *J* = 6.8 Hz, 6H); ^13^C NMR (101 MHz, CDCl_3_): δ 167.7, 165.9, 148.8, 139.3, 131.2, 129.1, 128.3, 126.4, 125.4, 123.7, 52.4, 32.9, 22.4; MS (*m*/*z*): 229 (M^+^); Anal. Calcd for C_14_H_15_NO_2_: C, 73.34; H, 6.59; N, 6.11. Found: C, 73.43; H, 6.55; N, 5.98.

#### 3.3.6. Methyl 2-(*tert*-Butyl)quinoline-3-carboxylate

Yield: 0.29 g (90%) as a light yellow oil; IR: 1728 cm^−1^; ^1^H NMR (400 MHz, CDCl_3_): δ 8.16 (s, 1H), 8.05 (d, *J* = 8.5 Hz, 1H), 7.77 (d, *J* = 8.2 Hz, 1H), 7.72 (t, *J* = 8.2 Hz, 1H), 7.51 (t, *J* = 8.2 Hz, 1H), 3.97 (s, 3H), 1.52 (s, 9H); ^13^C NMR (101 MHz, CDCl_3_): δ 170.8, 164.5, 147.1, 137.2, 130.4, 129.4, 127.4, 126.6, 126.4, 124.8, 52.7, 39.9, 30.0; MS (*m*/*z*): 243 (M^+^); Anal. Calcd for C_15_H_17_NO_2_: C, 74.05; H, 7.04; N, 5.76. Found: C, 74.13; H, 6.99; N, 5.71.

#### 3.3.7. Ethyl 2-Phenylquinoline-3-carboxylate

Yield: 0.34 g (99%) as a light yellow oil; IR: 1717 cm^−1^; ^1^H NMR (400 MHz, CDCl_3_): δ 8.65 (s, 1H), 8.19 (d, *J* = 8.5 Hz, 1H), 7.92 (d, *J* = 8.2 Hz, 1H), 7.81 (t, *J* = 8.2 Hz, 1H), 7.65–7.58 (complex, 3H), 7.50–7.43 (complex, 3H), 4.19 (q, *J* = 7.1 Hz, 2H), 1.07 (q, *J* = 7.1 Hz, 3H); ^13^C NMR (101 MHz, CDCl_3_): δ 168.0, 158.2, 148.4, 140.8, 139.1, 131.6, 129.6, 128.61, 128.58, 129.3, 128.2, 127.3, 125.9, 125.6, 61.6, 13.7; MS (*m*/*z*): 277 (M^+^). The spectra matched those reported in the literature [37].

#### 3.3.8. Methyl 2-Benzylquinoline-3-carboxylate

Yield: 0.37 g (99%) as a light yellow solid, m.p. 63–65 °C; IR: 1740 cm^−1^; ^1^H NMR (400 MHz, CDCl_3_): δ 8.69 (s, 1H), 8.12 (d, *J* = 8.5 Hz, 1H), 7.86 (d, *J* = 8.2 Hz, 1H), 7.80 (t, *J* = 8.2 Hz, 1H), 7.56 (t, *J* = 7.5 Hz, 1H), 7.27–7.18 (complex, 4H), 7.15 (m, 1H), 4.76 (s, 2H), 3.85 (s, 3H); ^13^C NMR (101 MHz, CDCl_3_): δ 166.9, 159.8, 148.6, 140.3, 139.5, 131.7, 129.04, 129.01, 128.4, 128.2, 126.9, 126.1, 125.9, 123.9, 52.4, 43.4; MS (*m*/*z*): 277 (M^+^); Anal. Calcd for C_18_H_15_NO_2_: C, 77.96; H, 5.45; N, 5.05. Found: C, 78.04; H, 5.48; N, 4.97.

#### 3.3.9. Methyl 2-Phenethylquinoline-3-carboxylate

Yield: 0.36 g (93%) as a light yellow solid, m.p. 58–59 °C (lit. [38] m.p. 58–59 °C); IR: 1724 cm^−1^; ^1^H NMR (400 MHz, CDCl_3_): δ 8.74 (s, 1H), 8.10 (d, *J* = 8.5 Hz, 1H), 7.87 (d, *J* = 8.1 Hz, 1H), 7.81 (t, *J* = 8.1 Hz, 1H), 7.56 (t, *J* = 8.1 Hz, 1H), 7.37–7.27 (complex, 4H), 7.21 (m, 1H), 3.97 (s, 3H), 3.65 (t, *J* = 8.2 Hz, 2H), 3.13 (t, *J* = 8.2 Hz, 2H); ^13^C NMR (101 MHz, CDCl_3_): δ 166.9. 161.0, 148.8, 142.0, 140.2, 131.7, 128.8, 128.7, 128.5, 128.3, 126.7, 125.9, 125.8, 123.6, 52.5, 39.7, 36.0; MS (*m*/*z*): 291 (M^+^). The spectra matched those reported elsewhere [38].

#### 3.3.10. Methyl 2-(Phenoxymethyl)quinoline-3-carboxylate

Yield: 0.37 g (96%) as a light yellow oil; IR: 1725 cm^−1^; ^1^H NMR (400 MHz, CDCl_3_): δ 8.73 (s, 1H), 8.14 (d, *J* = 8.5 Hz, 1H), 7.90 (d, *J* = 8.2 Hz, 1H), 7.82 (t, *J* = 8.2 Hz, 1H), 7.61 (t, *J* = 8.2 Hz, 1H), 7.28 (m, 2H), 7.02 (d, *J* = 7.7 Hz, 2H), 6.95 (t, *J* = 7.7 Hz, 1H), 5.66 (s, 2H), 3.89 (s, 3H); ^13^C NMR (101 MHz, CDCl_3_): δ 166.9, 158.9, 155.6, 148.2, 140.0, 131.8, 129.8, 129.4, 128.5, 127.6, 126.6, 123.9, 121.1, 114.9, 71.0, 52.6; MS (*m*/*z*): 293 (M^+^); Anal. Calcd for C_18_H_15_NO_3_: C, 73.71; H, 5.15; N, 4.78. Found: C, 73.56; H, 5.11; N, 4.70.

#### 3.3.11. 2-Methyl-3-(phenylsulfonyl)quinoline

Yield: 0.93 g (99%) as a light yellow solid, m.p. 142–144 °C (lit. [39] m.p. 145.5–146.5 °C); IR: 1592, 1563, 1315, 1156 cm^−1^; ^1^H NMR (400 MHz, CDCl_3_): δ 9.07 (s, 1H), 8.05 (d, *J* = 8.1 Hz, 1H), 7.99 (d, *J* = 8.1 Hz, 1H), 7.93 (apparent d, *J* = 8.2 Hz, 2H), 7.86 (t, *J* = 8.2 Hz, 1H), 7.67–7.60 (complex 2H), 7.54 (apparent t, *J* = 8.0 Hz, 2H), 2.79 (s, 3H); ^13^C NMR (101 MHz, CDCl_3_): δ 155.3, 149.2, 140.2, 139.4, 133.6, 133.5, 132.8, 129.3, 129.0, 128.6, 128.0, 127.5, 125.6, 24.2; MS (*m*/*z*): 283 (M^+^).

#### 3.3.12. 2-Methylquinoline-3-carbonitrile

Yield: 0.16 g (68%) as a white solid, m.p. 130–131 °C (lit. [40] m.p. 132–133.5 °C); IR: 2223 cm^−1^; ^1^H NMR (400 MHz, CDCl_3_): δ 9.01 (s, 1H), 8.04 (apparent t, *J* = 8.1 Hz, 2H), 7.92 (t, *J* = 8.1 Hz, 1H), 7.70 (t, *J* = 8.1 Hz, 1H), 2.82 (s, 3H); ^13^C NMR (101 MHz, CDCl_3_): δ 158.1, 148.1, 143.7, 133.4, 128.9, 128.8, 127.9, 125.0, 117.9, 106.7, 24.3; MS (*m*/*z*): 168 (M^+^).

#### 3.3.13. 2-Phenylquinoline-3-carbonitrile

Yield: 0.21 g (70%) as a white solid, m.p. 189–190 °C (lit. [41] m.p. 186 °C); IR: 2223 cm^−1^; ^1^H NMR (400 MHz, CDCl_3_): δ 8.68 (s, 1H), 8.20 (d, *J* = 8.0 Hz, 1H), 8.00 (m, 2H), 7.91 (m, 2H), 7.67 (t, *J* = 8.0 Hz, 1H), 7.61–7.54 (complex, 3H); ^13^C NMR (101 MHz, CDCl_3_): δ 158.1, 148.7, 144.2, 137.7, 133.0, 130.1, 130.0, 129.2, 128.8, 128.1, 127.8, 125.0, 118.0, 105.6; MS (*m*/*z*): 230 (M^+^).

#### 3.3.14. 3,3-Dimethyl-3,4-dihydroacridine-1(2*H*)-one

Yield: 0.30 g (99%) as a white solid, m.p. 109–110 °C (lit. [36] m.p. 101–102 °C); IR: 1690 cm^−1^; ^1^H NMR (400 MHz, DMSO-*d*_6_): δ 8.88 (s, 1H), 8.19 (d, *J* = 8.3 Hz, 1H), 8.01 (d, *J* = 8.3 Hz, 1H), 7.88 (t, *J* = 8.3 Hz, 1H), 7.63 (t, *J* = 8.3 Hz, 1H), 3.16 (s, 2H), 2.68 (s, 2H), 1.07 (s, 6H); ^13^C NMR (101 MHz, DMSO-*d*_6_): δ 198.0, 161.4, 149.8, 136.2, 132.8, 130.6, 128.5, 127.1, 126.8, 125.5, 54.1, 46.7, 32.9, 28.4; MS (*m*/*z*): 225 (M^+^).

#### 3.3.15. 1-(2-Methylquinolin-3-yl)ethan-1-one

Yield: 0.20 g (79%) as a light yellow solid, m.p. 73–75 °C (lit. [36] m.p. 72–74 °C); IR: 1672 cm^−1^; ^1^H NMR (400 MHz, DMSO-*d*_6_): δ 8.92 (s, 1H), 8.07 (d, *J* = 8.2, 1H), 7.96 (d, *J* = 8.2, 1H), 7.85 (t, *J* = 8.2 Hz, 1H), 7.64 (t, *J* = 8.2, 1H), 2.78 (s, 3H), 2.71 (s, 3H); ^13^C NMR (101 MHz, DMSO-*d*_6_): δ 201.0, 157.1, 148.0, 139.1, 132.2, 131.3, 129.3, 128.4, 127.1, 126.0, 29.8, 25.6; MS (*m*/*z*): 185 (M^+^).

#### 3.3.16. (2-Methylquinolin-3-yl)(phenyl)methanone

Yield: 0.28 g (86%) as a yellow oil; IR: 1662 cm^−1^; ^1^H NMR (400 MHz, DMSO-*d*_6_): δ 8.41 (s, 1H), 8.04 (d, *J* = 7.6 Hz, 2H), 7.88–7.83 (complex, 3H), 7.75 (t, *J* = 8.1 Hz, 1H), 7.66–7.54 (complex, 3H), 2.63 (s, 3H); ^13^C NMR (101 MHz, DMSO-*d*_6_): δ 196.8, 156.2, 147.9, 137.24, 137.20, 134.4, 132.2, 131.6, 130.4, 129.5, 129.1, 128.6, 127.1, 125.5, 24.2; MS (*m*/*z*): 247(M^+^); Anal. Calcd for C_17_H_13_NO: C, 82.57; H, 5.30; N, 5.66. Found: C, 82.46; H, 5.34; N, 5.54.

#### 3.3.17. 2-Methyl-3-phenylquinoline

Yield: 0.23 g (79%) as a light yellow oil; IR: 1380, 1350 cm^−1^; ^1^H NMR (400 MHz, CDCl_3_): δ 8.07 (d, *J* = 8.4 Hz, 1H), 7.94 (s, 1H), 7.77 (d, *J* = 8.3 Hz, 1H), 7.68 (t, *J* = 8.4 Hz, 1H), 7.52–7.38 (complex, 6H), 2.67 (s, 3H); MS (*m*/*z*): 219 (M^+^); ^13^C NMR (101 MHz, CDCl_3_): δ 157.4, 147.1, 140.0, 136.1, 135.8, 129.4, 129.2, 128.5 (2C), 127.6, 127.5, 126.9, 126.1, 24.7; MS (*m*/*z*): 219 (M^+^); Anal. Calcd for C_16_H_13_N: C, 87.64; H, 5.98; N, 6.39. Found: C, 87.44; H, 5.92; N, 6.30.

#### 3.3.18. 2,3-Diphenylquinoline

Yield: 0.23 g (61%) as a light yellow solid, m.p. 86–88 °C (lit. [42] m.p. 88–89 °C); IR: 1605, 1586, 1457 cm^−1^; ^1^H NMR (400 MHz, DMSO-*d*_6_): δ 8.41 (s, 1H), 8.08 (m, 2H), 7.91 (t, *J* = 8.2 Hz, 1H), 7.65 (t, *J* = 8.2 Hz, 1H), 7.38 (m, 2H), 7.34–7.24 (complex, 8 H); ^13^C NMR (101 MHz, DMSO-*d*_6_): δ 158.4, 147.3, 140.5, 140.0, 137.6, 134.6, 130.0, 129.8, 129.6, 129.5, 128.3, 128.02, 127.96, 127.5, 127.3, 127.2, 126.8; MS (*m*/*z*): 281 (M^+^).

#### 3.3.19. 5,6-Dihydrobenzo[*a*]acridine

Yield: 0.22 g (73%) as a tan solid, m.p. 84–86 °C (lit. [43] m.p. 86–87 °C); IR: 1495, 1406 cm^−1^; ^1^H NMR (400 MHz, CDCl_3_): δ 8.41 (d, *J* = 8.4 Hz, 1H), 8.04 (d, *J* = 8.4 Hz, 1H), 7.90 (d, *J* = 7.8 Hz, 1H), 7.86 (d, *J* = 8.1 Hz, 1H), 7.68 (t, *J* = 8.4 Hz, 1H), 7.51 (t, *J* = 8.2 Hz, 1H), 7.38 (m, 1H), 7.32 (m, 2H), 3.29 (m, 2H), 3.10 (m, 2H); ^13^C NMR (101 MHz, CDCl_3_): δ 159.4, 147.0, 137.5, 133.0, 129.6, 129.3, 128.5, 128.2, 128.0, 127.9, 127.3, 126.1, 124.4, 32.9, 28.8 (two aromatic C unresolved); MS (*m*/*z*): 231 (M^+^).

### 3.4. Reactions with 5-Fluoro-2-nitrobenzaldehydes (***1b***)

#### 3.4.1. Ethyl 6-Fluoro-2-(trifluoromethyl)quinoline-3-carboxylate

Yield: 0.24 g (70%) as a light yellow solid, m.p. 81–82 °C; 1716, 1214, 1141, 1117 cm^−1^; ^1^H NMR (400 MHz, DMSO-*d*_6_): δ 9.05 (s, 1H), 8.34 (dd, *J* = 9.3, 5.3 Hz, 1H), 8.12 (dd, *J* = 9.0, 2.9 Hz, 1H), 7.98 (td, *J* = 8.9, 2.9 Hz, 1H), 4.43 (q, *J* = 7.1 Hz, 2H), 1.37 (t, *J* = 7.1 Hz, 3H); ^13^C NMR (101 MHz, DMSO-*d*_6_): δ 165.1, 162.0 (d, *J* = 250.6 Hz), 142.9, 143.0 (q, *J* = 37.6 Hz), 140.6 (d, *J* = 5.6 Hz), 133.0 (d, *J* = 9.8 Hz), 129.1 (d, *J* = 11.2 Hz), 124.5, 123.9 (d, *J* = 26.3 Hz), 121.5 (q, *J* = 275.5 Hz), 112.6 (d, *J* = 23.0 Hz), 62.8, 14.3; MS (*m*/*z*): 287 (M^+^); Anal. Calcd for C_13_H_9_F_4_NO_2_: C, 54.36; H, 3.16; N, 4.88. Found: C, 54.29; H, 3.19; N, 4.76. 

#### 3.4.2. Methyl 6-Fluoro-2-isopropylquinoline-3-carboxylate

Yield: 0.20 g (67%) as a white solid, m.p. 72–73 °C; IR: 1727, 1265, 1218 cm^−1^; ^1^H NMR (400 MHz, DMSO-*d*_6_): δ 8.77 (s, 1H), 8.07 (dd, *J* = 9.3, 5.4 Hz. 1H), 7.91 (d, *J* = 9.3 Hz, 1H), 7.78 (t, *J* = 8.9 Hz, 1H), 3.94 (s, 3H), 3.88 (septet, *J* = 6.7 Hz, 1H), 1.31 (d, *J* = 6.7 Hz, 6H); ^13^C NMR (101 MHz, DMSO-*d*_6_): δ 167.3, 164.6 (d, *J* = 2.7 Hz), 160.1 (d, *J* = 235.7 Hz), 145.3, 139.0 (d, *J* = 5.5 Hz), 131.7 (d, *J* = 9.2 Hz), 126.3 (d, *J* = 10.7 Hz), 124.9 122.1 (d, *J* = 26.0 Hz), 112.1 (d, *J* = 22.1 Hz), 53.1, 32.7, 22.8; MS (*m*/*z*): 247 (M^+^); Anal. Calcd for C_14_H_14_FNO_2_: C, 68.00; H, 5.71; N, 5.66. Found: C, 67.91; H, 5.68; N, 5.55.

#### 3.4.3. Ethyl 6-Fluoro-2-phenylquinoline-3-carboxylate

Yield: 0.29 g (82%) as a white solid, m.p. 82–83 °C; IR: 1707, 1258, 1210 cm^−1^; ^1^H NMR (400 MHz, DMSO-*d*_6_): δ 8.84 (s, 1H), 8.18 (dd, *J* = 9.3, 5.3 Hz, 1H), 8.00 (dd, *J* = 9.2, 2.9 Hz, 1H), 7.83 (td, *J* = 8.9, 2.9 Hz, 1H), 7.61–7.58 (complex, 2H), 7.53–7.48 (complex, 3H), 4.18 (q, *J* = 7.1 Hz, 2H), 1.06 (t, *J* = 7.1 Hz, 3H); ^13^C NMR (101 MHz, DMSO-*d*_6_): δ 167.7, 160.5 (d, *J* = 246.7 Hz), 156.8 (d, *J* = 2.6 Hz), 145.3, 140.3, 138.7 (d, *J* = 5.5 Hz), 132.0 (d, *J* = 9.5 Hz), 129.2, 129.0, 128.6, 126.7 (d, *J* = 11.0 Hz), 126.5, 122.5 (d, *J* = 26.0 Hz), 112.2 (d, *J* = 22.1 Hz), 61.9, 14.0; MS (*m*/*z*): 295 (M^+^); Anal. Calcd for C_18_H_14_FNO_2_: C, 73.21; H, 4.78; N, 4.74. Found: C, 73.06; H, 4.72; N, 4.65.

#### 3.4.4. Methyl 2-Benzyl-6-fluoroquinoline-3-carboxylate

Yield: 0.30 g (85%) as a white solid, m.p. 84–85 °C; IR: 1730, 1276, 1210 cm^−1^; ^1^H NMR (400 MHz, DMSO-*d*_6_): δ 8.85 (s, 1H), 8.12 (dd, *J* = 9.2, 5.3 Hz, 1H), 7.95 (dd, *J* = 9.2, 2.9 Hz, 1H), 7.80 (td, *J* = 8.9, 2.9 Hz, 1H), 7.28–7.22 (complex, 2H), 7.19–7.14 (complex, 3H), 4.61 (s, 2H), 3.84 (s, 3H); ^13^C NMR (101 MHz, DMSO-*d*_6_): δ 166.9, 160.3 (d, *J* = 246.2 Hz), 158.8 (d, *J* = 2.7 Hz), 145.6, 139.9 (d, *J* = 5.4 Hz), 139.7, 131.7 (d, *J* = 9.3 Hz), 129.2, 128.7, 126.7, (d, *J* = 10.7 Hz), 126.6, 124.9, 122.5 (d, *J* = 25.9 Hz), 112.3 (d, *J* = 22.1 Hz), 53.0, 42.7; MS (*m*/*z*): 295 (M^+^); Anal. Calcd for C_18_H_14_FNO_2_: C, 73.21; H, 4.78; N, 4.74. Found: C, 73.15; H, 4.76; N, 4.69.

#### 3.4.5. 6-Fluoro-2-Methyl-3-(phenylsulfonyl)quinoline

Yield: 0.20 g (80%) as a light yellow solid, m.p. 157–158 °C; IR: 1322, 1151 cm^−1^; ^1^H NMR (400 MHz, DMSO-*d*_6_): δ 9.27 (s, 1H), 8.15 (dd, *J* = 9.2, 2.9 Hz, 1H), 8.10 (dd, *J* = 9.2, 5.3 Hz, 1H), 7.97 (d, *J* = 7.6 Hz, 2H), 7.89 (td, *J* = 9.0, 6.0 Hz, 1H), 7.77 (t, *J* = 7.4 Hz, 1H), 7.68 (t, *J* = 7.4 Hz, 2H), 2.65 (s, 3H); ^13^C NMR (101 MHz, DMSO-*d*_6_): δ 160.6 (d, *J* = 247.1 Hz), 154.0 (d, *J* = 2.8 Hz), 146.2, 140.0, 139.7 (d, *J* = 5.5 Hz), 134.7, 134.1, 131.4 (d, *J* = 9.5 Hz), 130.3, 128.1, 126.7, (d, *J* = 11.0 Hz), 123.6 (d, *J* = 25.9 Hz), 113.3,(d, *J* = 22.4 Hz), 24.1; MS (*m*/*z*): 301; Anal. Calcd for C_16_H_12_NO_2_S: C, 63.77; H, 4.01; N, 4.65. Found: C, 63.81; H, 4.07; N, 4.49. 

#### 3.4.6. 6-Fluoro-2-phenylquinoline-3-carbonitrile

Yield: 0.20 g (68%) as a light yellow foam, m.p. 213–214 °C; IR: 2219, 1217, 1153 cm^−1^; ^1^H NMR (400 MHz, CDCl_3_): δ 8.62 (s, 1H), 8.23 (dd, *J* = 9.2, 5.1 Hz, 1H), 7.99 (m, 2H), 7.67 (td, *J* = 9.2, 2.8 Hz, 1H), 7.61–7.51 (complex, 4H); ^13^C NMR (101 MHz, CDCl_3_): δ 161.1(d, *J* = 251.9 Hz), 157.4 (*J* = 2.9 Hz), 145.9, 143.4 (d, *J* = 5.8 Hz), 137.4, 132.6 (d, *J* = 9.2 Hz), 130.2, 129.1, 128.8, 125.7 (d, *J* = 10.5 Hz), 123.3 (d, *J* = 25.8 Hz), 117.6, 110.8 (d, *J* = 22.3 Hz), 106.6; MS (*m*/*z*): 248 (M^+^); Anal. Calcd for C_16_H_9_FN_2_: C, 77.41; H, 3.65; N, 11.28. Found: C, 77.32; H, 3.71; N, 11.20.

#### 3.4.7. 7-Fluoro-3,3-dimethyl-3,4-dihydroacridine-1(2*H*)-one

Yield: 0.23 g (80%) as a light yellow solid, m.p. 146–147 °C; IR: 1691, 1198 cm^−1^; ^1^H NMR (400 MHz, DMSO-*d*_6_): δ 8.88 (s, 1H), 8.06 (dd, *J* = 9.3, 5.3 Hz, 1H), 8.02 (dd, *J* = 9.3, 2.9 Hz, 1H), 7.80 (td, *J* = 8.9, 2.9 Hz, 1H), 3.15 (s, 2H), 2.68 (s, 2H), 1.06 (s, 6H); ^13^C NMR (101 MHz, DMSO-*d*_6_): δ 197.9, 160.8 (d, *J* = 2.6 Hz), 160.0 (d, *J* = 245.7 Hz), 147.0, 135.8 (d, *J* = 5.7 Hz), 131.4 (d, *J* = 9.2 Hz), 127.5 (d, *J* = 10.7 Hz), 125.9, 122.8 (d, *J* = 26.4 Hz), 113.3 (d, *J* = 21.8 Hz), 52.1, 46.5, 32.8, 28.3; MS (*m*/*z*): 243 (M^+^); Anal. Calcd for C_15_H_14_FNO: C, 74.06; H, 5.80; N, 5.76. Found: C, 73.99; H, 5.74; N, 5.68.

### 3.5. Reactions with 5-Methoxy-2-nitrobenzaldehyde (***1c***)

#### 3.5.1. Ethyl 6-Methoxy-2-(trifluoromethyl)quinoline-3-carboxylate

Yield: 0.23 g (65%) as a light yellow solid, m.p. 59–60 °C; IR: 2849, 1730, 1170, 1116 cm^−1^; ^1^H NMR (400 MHz, DMSO-*d*_6_): δ 8.87 (s, 1H), 8.12 (d, *J* = 8.9 Hz, 1H), 7.65 (overlapping dd, *J* = 8.9, 2.6 Hz, 1H and s, 1H), 4.42 (q, *J* = 7.1 Hz, 2H), 3.97 (s, 3H), 1.38 (t, *J* = 7.1 Hz, 3H); ^13^C NMR (101 MHz, DMSO-*d*_6_): δ 165.4, 160.2, 142.6, 140.9 (q, *J* = 34.4 Hz), 139.2, 131.2, 129.5, 126.2, 124.1, 121.8 (q, *J* = 275.0 Hz), 106.7, 62.6, 56.4, 14.3; MS (*m*/*z*): 299 (M^+^); Anal. Calcd for C_14_H_12_F_3_NO_3_: C, 56.19; H, 4.04; N, 4.68. Found: C, 56.22 H, 4.07; N, 4.59.

#### 3.5.2. Methyl 2-Isopropyl-6-methoxyquinoline-3-carboxylate

Yield: 0.21 g (68%) as a yellow oil; IR: 2847, 1728 cm^−1^; ^1^H NMR (400 MHz, CDCl_3_): δ 8.65 (s, 1H), 7.91 (d, *J* = 9.9 Hz, 1H), 7.48 (m, 2H), 3.94 (s, 3H), 3.91 (s, 3H), 3.85 (septet, *J* = 6.7 Hz, 1H), 1.31 (d, *J* = 6.7 Hz, 6H); ^13^C NMR (101 MHz, CDCl_3_): δ 167.6, 162.5, 157.7, 144.5, 138.2, 130.3, 126.7, 124.5, 124.1, 106.7, 56.0, 53.0, 34.2, 22.8; MS (*m*/*z*): 259 (M^+^); Anal. Calcd for C_15_H_17_NO_3_: C, 69.48; H, 6.61; N, 5.40. Found: C, 69.42; H, 6.59; N, 5.29.

#### 3.5.3. Ethyl 6-Methoxy-2-phenylquinoline-3-carboxylate

Yield: 0.27 g (80%) as a white solid, m.p. 109–110 °C; IR: 2839, 1719 cm^−1^; ^1^H NMR (400 MHz, CDCl_3_): δ 8.70 (s, 1H), 8.01 (d, *J* = 9.1 Hz, 1H), 7.59–7.45 (complex, 7H), 4.17 (q, *J* = 7.1 Hz, 2H), 3.94 (s, 3H), 1.07 (t, *J* = 7.1 Hz, 3H); ^13^C NMR (101 MHz, CDCl_3_): δ 168.0, 158.4, 154.8, 144.2, 140.6, 137.8, 130.8, 129.0, 128.8, 128.5, 127.2, 125.9, 124.8, 106.6, 61.7, 56.2, 14.0; MS (*m*/*z*): 307 (M^+^); Anal. Calcd for C_19_H_17_NO_3_: C, 74.25; H, 5.58; N, 4.56. Found: C, 74.23; H, 5.54; N, 4.49.

#### 3.5.4. Methyl 2-Benzyl-6-methoxyquinoline-3-carboxylate

Yield: 0.28 g (80%) as a light yellow solid, m.p. 93–95 °C; IR: 2845, 1730 cm^−1^; ^1^H NMR (400 MHz, DMSO-*d*_6_): δ 8.74 (s, 1H), 7.95 (d, *J* = 8.9 Hz, 1H), 7.51 (overlapping s, 1H and dd, *J* = 8.9, 2.6 Hz, 1H), 7.26–7.21 (complex, 2H), 7.19–7.12 (complex, 3H), 4.58 (s, 2H), 3.90 (s, 3H), 3.83 (s, 3H); ^13^C NMR (101 MHz, DMSO-*d*_6_): δ 167.1, 158.0, 156.7, 144.6, 140.1, 139.1, 130.3, 129.1, 128.7, 127.1, 126.5, 124.9, 124.3, 106.9, 56.1, 52.9, 42.6; MS (*m*/*z*): 307 (M^+^); Anal. Calcd for C_19_H_17_NO_3_: C, 74.25; H, 5.58; N, 4.56. Found: C, 74.17; H, 5.51; N, 4.54.

#### 3.5.5. 6-Methoxy-2-methyl-3-(phenylsulfonyl)quinoline

Yield: 0.36 g (82%) as a white solid, m.p. 194–195 °C; IR: 2839, 1619, 1584, 1312, 1153 cm^−1^; ^1^H NMR (400 MHz, DMSO-*d*_6_): δ 9.11 (s, 1H), 7.96 (d, *J* = 8.1 Hz, 2H), 7.92 (d, *J* = 9.1 Hz, 1H), 7.76 (t, *J* = 8.1 Hz, 1H), 7.73 (d, *J* = 2.9 Hz, 1H), 7.68 (t, *J* = 8.1 Hz, 2H), 7.58 (dd, *J* = 9.1, 2.9 Hz, 1H), 3.94 (s, 3H), 2.63 (s, 3H); ^13^C NMR (101 MHz, DMSO-*d*_6_): δ 158.3, 151.7, 145.1, 140.3, 138.6, 134.5, 133.6, 130.3, 129.9, 128.0, 127.1, 126.0, 107.7, 55.3, 23.9; MS (*m*/*z*): 313 (M^+^); Anal. Calcd for C_17_H_15_NO_3_S: C, 65.16; H, 4.82; N, 4.47. Found: C, 65.04; H, 4.77; N, 4.41.

#### 3.5.6. 6-Methoxy-2-phenylquinoline-3-carbonitrile

Yield: 0.22 g (74%) as a light yellow foam, m.p. 166–168 °C (lit. [44] m.p. 170 °C); IR: 2841, 2224 cm^−1^; ^1^H NMR (400 MHz, CDCl_3_): δ 8.54 (s, 1H), 8.10 (d, *J* = 9.3 Hz, 1H), 7.98 (m, 2H), 7.57–7.50 (complex, 4H), 7.13 (d, *J* = 2.8 Hz, 1H), 3.98 (s, 3H); ^13^C NMR (101 MHz, CDCl_3_): δ 158.9, 155.8, 145.0, 142.5, 137.8, 131.4, 129.8, 129.0, 128.7, 126.2, 126.1, 118.2, 105.7, 104.6, 55.8; MS (*m*/*z*): 260 (M^+^).

#### 3.5.7. 7-Methoxy-3,3-dimethyl-3,4-dihydroacridine-1(2*H*)-one

Yield: 0.21 g (75%) as a light yellow solid, m.p. 152–153 °C; IR: 2838, 1684 cm^−1^; ^1^H NMR (400 MHz, DMSO-*d*_6_): δ 8.75 (s, 1H), 7.91 (d, *J* = 9.2 Hz, 1H), 7.59 (d, *J* = 2.9 Hz, 1H), 7.51 (dd, *J* = 9.2, 2.9 Hz, 1H), 3.90 (s, 3H), 3.11 (s, 2H), 2.65 (s, 2H), 1.06 (s, 6H); ^13^C NMR (101 MHz, DMSO-*d*_6_): δ 198.1, 158.7, 157.6, 146.0, 134.7, 130.0, 127.9, 125.5, 125.4, 107.8, 56.1, 52.2, 46.4, 32.9, 28.4; MS (*m*/*z*): 262 (M^+^); Anal. Calcd for C_16_H_17_NO_2_: C, 75.27; H, 6.71; N, 5.49. Found: C, 75.18; H, 6.67; N, 5.39.

### 3.6. Reactions with 2-Nitroacetophenone (***1d***)

#### 3.6.1. 4-Methyl-3-propionylquinolin-2(1*H*)-one

Yield: 0.23 g (83%) as a white solid, m.p. 192–193 °C; IR: 3305, 1668 cm^−1^; ^1^H NMR (400 MHz, CDCl_3_): δ 11.6 (br s, 1H), 7.76 (dd, *J* = 8.2, 1.3 Hz, 1H), 7.55 (ddd, *J* = 8.2, 7.1, 1.3 Hz, 1H), 7.32 (d, *J* = 8.2 Hz, 1H), 7.27 (ddd, *J* = 8.2, 7.1, 1.2 Hz, 1H), 2.96 (q, *J* = 7.3 Hz, 2H), 2.44 (s, 3H), 1.25 (t, *J* = 7.3 Hz, 3H); ^13^C NMR (101 MHz, CDCl_3_): δ 206.7, 161.5, 145.4, 137.7, 132.7, 131.2, 125.3, 123.0, 120.2, 116.4, 37.3, 15.8, 7.9; MS (*m*/*z*): 215 (M^+^); Anal. Calcd for C_13_H_13_NO_2_: C, 72.54; H, 6.09; N, 6.51. Found: C, 72.46; H, 6.01; N, 6.50.

#### 3.6.2. 3-Benzoyl-4-methylquinolin-2(1*H*)-one

Yield: 0.29 g (93%) as a white solid, m.p. 260–261 °C (lit. [45] m.p. 262–264 °C); IR: 3260, 1636 cm^−1^; ^1^H NMR (400 MHz, CDCl_3_): δ 12.0 (br s, 1H), 7.85 (apparent t, *J*~7.5 Hz, 3H), 7.68 (t, *J* = 7.6 Hz, 1H), 7.60 (t, *J* = 8.3 Hz, 1H), 7.54 (t, *J* = 7.6 Hz, 2H), 7.39 (d, *J* = 8.3 Hz, 1H), 7.29 (d, *J* = 8.3 Hz, 1H), 2.28 (s, 3H); ^13^C NMR (101 MHz, CDCl_3_): δ 196.2, 160.2, 145.1, 138.8, 136.9, 134.4, 131.6, 131.3, 129.5, 129.4, 125.9, 122.7, 119.6, 116.2, 16.1; MS (*m*/*z*): 263 (M^+^).

#### 3.6.3. 4-Methyl-2-phenylquinoline-3-carbonitrile

Yield: 0.20 g (68%) as a light yellow solid, m.p. 164–165 °C (lit. [46] m.p. 156–157 °C); IR: 1678 cm^−1^; ^1^H NMR (400 MHz, CDCl_3_): δ 8.19 (d, *J* = 8.3 Hz, 1H), 8.11 (d, *J* = 8.3 Hz, 1H), 7.96–7.92 (complex, 2H), 7.87 (ddd, *J* = 8.3, 6.8, 1.4 Hz, 1H), 7.68 (ddd, *J* = 8.3, 6.8, 1.3 Hz, 1H), 7.59–7.50 (complex, 3H), 3.05 (s, 3H); ^13^C NMR (101 MHz, CDCl_3_): δ 158.4, 152.9, 147.9, 138.3, 132.4, 130.6, 129.9, 129.2, 128.6, 127.8, 125.2, 124.3, 117.4, 106.4, 17.8; MS (*m*/*z*): 244 (M^+^). Anal. Calcd for C_17_H_12_N_2_: C, 83.58; H, 4.95; N, 11.47. Found: C, 83.49; H, 4.98; N, 11.39.

#### 3.6.4. 1-(2,4-Dimethylquinolin-3-yl)ethan-1-one

Yield: 0.15 g (63%) as a light yellow oil; IR: 1704 cm^−1^; ^1^H NMR (400 MHz, CDCl_3_): δ 8.02 (d, *J* = 8.4 Hz, 1H), 7.97 (d, *J* = 8.4 Hz, 1H), 7.71 (t, *J* = 8.4 Hz, 1H), 7.55 (t, *J* = 8.4 Hz, 1H), 2.63 (s, 3H), 2.59 (s, 3H), 2.59 (s, 3H); ^13^C NMR (101 MHz, CDCl_3_): δ 206.7, 152.6, 147.0, 138.6, 135.7, 129.8, 129.3, 126.4, 126.0, 123.7, 32.7, 23.6, 15.2; MS (*m*/*z*): 199 (M^+^); Anal. Calcd for C_13_H_13_NO: C, 78.36; H, 6.58; N, 7.03. Found: C, 78.23; H, 6.55; N, 6.92.

#### 3.6.5. 3,3,9-Trimethyl-3,4-dihydroacridin-1(2*H*)-one

Yield: 0.23 g (79%) as a light yellow solid, m.p. 101–103 °C (lit. [47] m.p. 104–106 °C); IR: 1678 cm^−1^; ^1^H NMR (400 MHz, CDCl_3_): δ 8.22 (d, *J* = 8.4 Hz, 1H), 8.01 (d, *J* = 8.4 Hz, 1H), 7.77 (ddd, *J* = 8.3, 6.8, 1.4 Hz, 1H), 7.57 (ddd, *J* = 8.3, 6.8, 1.4 Hz, 1H), 3.19 (s, 2H), 3.07 (s, 3H), 2.67 (s, 2H), 1.14 (s, 6H); ^13^C NMR (101 MHz, CDCl_3_): δ 200.7, 161.1, 149.7, 148.3, 131.4, 129.2, 127.7, 126.4, 125.5, 124.2, 54.9, 48.6, 32.1, 28.3, 16.0; MS (*m*/*z*): 239 (M^+^); Anal. Calcd for C_16_H_17_NO: C, 80.30; H, 7.16; N, 5.85. Found: C, 80.09; H, 7.07; N, 5.77.

#### 3.6.6. 2,4-Dimethyl-3-phenylquinoline

Yield: 0.16 g (57%) as a yellow oil; IR: 1586, 1496, 1380 cm^−1^; ^1^H NMR (400 MHz, CDCl_3_): δ 8.06 (d, *J* = 8.4 Hz, 1H), 7.99 (t, *J* = 8.4 Hz, 1H), 7.68 (t, *J* = 8.3 Hz, 1H), 7.55–7.45 (complex, 3H), 7.42 (m, 1H), 7.21 (d, *J* = 8.0 Hz, 2H), 2.43 (s, 3H), 2.39 (s, 3H); ^13^C NMR: (100 MHz, CDCl_3_): δ 157.6, 146.6, 141.2, 139.5, 134.9, 129.3, 129.2, 128.9, 128.7, 127.4, 126.7, 125.8, 124.1, 25.4, 15.9; MS (*m*/*z*): 233 (M^+^). Anal. Calcd for C_17_H_15_N: C, 87.52; H, 6.48; N, 6.00. Found: C, 87.37; H, 6.41; N, 5.88.

#### 3.6.7. 12-Methyl-5,6-dihydrobenzo[*a*]acridine

Yield: 0.20 g (68%) as a yellow oil; IR: IR: 1590, 1563, 1504, 1451, 1372 cm^−1^; ^1^H NMR (400 MHz, CDCl_3_): δ 8.07 (dd, *J* = 8.4, 1.4 Hz, 1H), 8.02 (dd, *J* = 8.4, 1.4 Hz, 1H), 7.67 (ddd, *J* = 8.3, 6.9, 1.4 Hz, 1H), 7.61 (d, *J* = 7.7 Hz, 1H), 7.54 (ddd, *J* = 8.3, 5.5, 1.4 Hz, 1H), 7.38–7.26 (complex, 3H), 3.17 (m, 2H), 2.96 (m, 2H), 2.94 (s, 3H); ^13^C NMR: (100 MHz, CDCl_3_): δ 160.9, 145.9, 140.4, 139.0, 133.6, 130.1, 129.0, 128.8, 128.6, 127.8127.72, 127.70, 126.1, 125.8, 124.4, 34.6, 29.3, 17.4; MS (*m*/*z*): 245 (M^+^). Anal. Calcd for C_18_H_15_N: C, 88.13; H, 6.16; N, 5.71. Found: C, 87.95; H, 6.22; N, 5.65.

### 3.7. Reactions with 2-Nitrobenzophenone (***1e***)

#### 3.7.1. Methyl 2-Ethyl-4-phenylquinoline-3-carboxylate

Yield: 0.20 g (78%) as a light yellow solid, m.p. 102–104 °C (lit. [47] 105–106 °C); IR: 1738 cm^−1^; ^1^H NMR (400 MHz, CDCl_3_): δ 8.12 (d, *J* = 8.4 Hz, 1H), 7.72 (ddd, *J* = 8.4, 6.8, 1.5 Hz, 1H), 7.59 (dd, *J* = 8.4, 2.0 Hz, 1H), 7.53–7.46 (complex, 3H), 7.43 (ddd, J = 8.4, 6.8, 1.4 Hz, 1H), 7.37–7.34 (complex, 2H), 3.56 (s, 3H), 3.05 (q, *J* = 7.5 Hz, 2H), 1.43 (t, *J* = 7.5 Hz, 3H); ^13^C NMR (101 MHz, CDCl_3_): δ 169.1, 159.2, 147.9, 146.5, 135.8, 130.2, 129.3, 129.1, 128.5, 128.3, 127.0, 126.51, 126.45, 125.1, 52.1, 30.4, 13.8; MS (*m*/*z*): 291 (M^+^); Anal. Calcd for C_19_H_17_NO_2_: C, 78.33; H, 5.88; N, 4.81. Found: C, 78.22; H, 5.83; N, 4.69.

#### 3.7.2. 3-Benzoyl-4-phenylquinolin-2(1*H*)-one

Yield: 0.24 g (85%) from ethyl benzoylacetate; 0.23 g (82%) from benzoylacetonitrile as a white solid, m.p. 260–262 °C (lit. [48] m.p. 265–267 °C); IR: 3268, 1640 cm^−1^; ^1^H NMR (400 MHz, CDCl_3_): δ 12.5 (br s, 1H), 7.76 (d, *J* = 8.3 Hz, 2H), 7.59 (overlapping ddd, *J* = 8.4, 7.0, 1.5 Hz, 1H and t, *J* = 7.9 Hz, 1H), 7.47 (d, *J* = 8.3 Hz, 1H), 7.43 (t, *J* = 7.9 Hz, 2H), 7.38–7.34 (complex, 3H), 7.26–7.22 (complex, 2H), 7.17 (ddd, *J* = 8.3, 7.0, 1.2 Hz, 1H), 7.10 (dd, *J* = 8.3, 1.4 Hz, 1H); ^13^C NMR (101 MHz, CDCl_3_): δ 194.7, 160.2, 148.8, 139.3, 137.0, 134.3, 134.1, 131.7, 131.6, 129.4, 129.2, 129.1, 128.7, 127.4, 122.9, 119.5, 116.3 (one aromatic carbon unresolved); MS (*m*/*z*): 325 (M^+^).

#### 3.7.3. 3-Benzoyl-4-phenylquinolin-2(1*H*)-one

Yield: 0.24 g (85%) from ethyl benzoylacetate; 0.23 g (82%) from benzoylacetonitrile as a white solid, m.p. 260–262 °C (lit. [48] m.p. 265–267 °C); IR: 3268, 1640 cm^−1^; ^1^H NMR (400 MHz, CDCl_3_): δ 12.5 (br s, 1H), 7.76 (d, *J* = 8.3 Hz, 2H), 7.59 (overlapping ddd, *J* = 8.4, 7.0, 1.5 Hz, 1H and t, *J* = 7.9 Hz, 1H), 7.47 (d, *J* = 8.3 Hz, 1H), 7.43 (t, *J* = 7.9 Hz, 2H), 7.38–7.34 (complex, 3H), 7.26–7.22 (complex, 2H), 7.17 (ddd, *J* = 8.3, 7.0, 1.2 Hz, 1H), 7.10 (dd, *J* = 8.3, 1.4 Hz, 1H); ^13^C NMR (101 MHz, CDCl_3_): δ 194.7, 160.2, 148.8, 139.3, 137.0, 134.3, 134.1, 131.7, 131.6, 129.4, 129.2, 129.1, 128.7, 127.4, 122.9, 119.5, 116.3 (one aromatic carbon unresolved); MS (*m*/*z*): 325 (M^+^).

#### 3.7.4. (2-Methyl-4-phenylquinolin-3-yl)ethan-1-one

Yield: 0.20 g (88%) as a light yellow solid, m.p. 112–114 °C (lit. [49] m.p. 114–115 °C); IR: 1698 cm^−1^; ^1^H NMR (400 MHz, DMSO-*d*_6_): δ 8.04 (d, *J* = 8.3 Hz, 1H), 7.80 (t, *J* = 8.3 Hz, 1H), 7.60–7.54 (complex, 4H), 7.51 (t, *J* = 8.3 Hz, 1H), 7.36 (m, 2H), 2.61 (s, 3H), 2.04 (s, 3H); ^13^C NMR (100 MHz, DMSO-*d*_6_): δ 205.6, 153.6, 147.3, 143.7, 135.12, 135.07, 130.7, 130.2, 129.5, 129.2, 129.1, 127.4, 126.2, 125.0, 32.3, 23.9; MS (*m*/*z*): 261 (M^+^).

#### 3.7.5. 3,3-Dimethyl-9-phenyl-3,4-dihydroacridin-1(2*H*)-one

Yield: 0.23 g (85%) as a light yellow solid, m.p. 191–193 °C (lit. [49] m.p. 195 °C); IR: 1678 cm^−1^; ^1^H NMR (400 MHz, CDCl_3_): δ 8.07 (d, *J* = 8.4 Hz, 1H), 7.76 (ddd, *J* = 8.3, 6.7, 1.4 Hz, 1H), 7.56–7.46 (complex, 4H), 7.40 (ddd, *J* = 8.3, 6.7, 1.4 Hz, 1H), 7.20–7.16 (complex, 2H), 3.28 (s, 2H), 2.57 (s, 2H), 1.16 (s, 6H); ^13^C NMR (101 MHz, CDCl_3_): δ 198.0, 161.2, 151.0, 149.0, 137.6, 131.7, 128.5, 128.3, 128.13, 128.05, 127.5, 127.4, 126.4, 122.7, 54.2, 48.4, 32.3, 28.4; MS (*m*/*z*): 301 (M^+^).

#### 3.7.6. 2-Methyl-3,4-diphenylquinoline

Yield: 0.28 g (85%) as a white solid, m.p. 170–172 °C (lit. [50] m.p. 172–173 °C); IR: 1569, 1484, 1375 cm^−1^; ^1^H NMR (400 MHz, DMSO-*d*_6_): δ 8.04 (d, *J* = 8.4 Hz, 1H), 7.74 (t, *J* = 8.4 Hz, 1H), 7.48 (t, *J* = 8.4 Hz, 1H), 7.34 (d, *J* = 8.4 Hz, 1H), 7.32–7.22 (complex, 5H), 7.21–7.13 (complex, 5H), 2.42 (s, 3H); ^13^C NMR: (100 MHz, DMSO-*d*_6_): δ 157.6, 146.9, 146.3, 138.8, 136.9, 134.2, 130.34, 130.28, 129.6, 128.9, 128.4, 128.2, 127.8, 127.4, 126.6, 126.5, 126.2, 25.5; MS (*m*/*z*): 295 (M^+^).

#### 3.7.7. 3-Butyl-2,4-diphenylquinoline

Yield: 0.17 g (58%) as a yellow oil; IR: 1576, 1486, 1380 cm^−1^; ^1^H NMR (400 MHz, CDCl_3_): δ 8.16 (d, *J* = 8.4 Hz, 1H), 7.62 (ddd, *J* = 8.3, 6.2, 2.0 Hz, 1H), 7.57 (m, 2H), 7.54–7.37 (complex, 6H), 7.36–7.28 (complex, 4H), 2.56 (m, 2H), 1.15 (m, 2H), 0.89 (sextet, *J* = 7.3 Hz, 2H), 0.48 t, *J* = 7.3 Hz, 3H); ^13^C NMR: (100 MHz, CDCl_3_): δ 161.2, 147.5, 146.1, 141.7, 137.5, 132.0, 129.6, 129.4, 128.8, 128.6, 128.4, 128.3, 128.0, 127.8, 127.5, 126.23, 126.18, 32.6, 29.8, 22.5, 13.3; MS (*m*/*z*): 337 (M^+^). Anal. Calcd for C_25_H_23_N: C, 88.98; H, 6.87; N, 4.15. Found: C, 88.79; H, 6.92; N, 4.06.

#### 3.7.8. 12-Phenyl-5,6-dihydrobenzo[*a*]acridine

Yield: 0.21 g (68%) as a light yellow solid, m.p. 116–117 °C; IR: 1560, 1490, 1391 cm^−1^; ^1^H NMR (400 MHz, CDCl_3_): δ 8.06 (dd, *J* = 8.4, 1.3 Hz, 1H), 7.66 (ddd, *J* = 8.3, 6.8, 1.4 Hz, 1H), 7.60 (dd, *J* = 8.4, 1.4 Hz, 1H), 7.52–7.45 (complex, 3H), 7.38 (ddd, *J* = 8.3, 6.8, 1.4 Hz, 1H), 7.35–7.29 (complex, 2H), 7.25 (obscured d, 1H), 7.09 (m, 1H), 6.82 (m 2H), 3.27 (m, 2H), 3.02 (m, 2H); ^13^C NMR: (100 MHz, CDCl_3_): δ 161.0, 146.5, 143.9, 140.0, 138.0, 132.9, 130.4, 130.0, 129.1, 128.9, 128.6, 128.0, 127.7, 127.6, 127.4, 126.6, 126.1, 125.9, 125.7, 34.7, 29.4; MS (*m*/*z*): 307 (M^+^). Anal. Calcd for C_23_H_17_N: C, 89.87; H, 5.57; N, 4.56. Found: C, 89.58; H, 5.78; N, 4.47.

## 4. Conclusions

The current work aimed to expand the scope of the Friedlander synthesis of quinolines by using an in situ dissolving metal reduction of more plentiful 2-nitroaromatic substrates in the reaction. The reaction proceeded in high yield and provided clean products from 2-nitrobenzaldehydes. Results with 2-nitroacetophenones and 2-nitrobenzophenones with ketones and β-diketones also provided excellent yields of highly substituted quinolines. However, 2-nitroaromatic ketones with β-keto-esters and β-keto-nitriles led to competitive cyclizations of the *Z* double bond isomer of the Knoevenagel intermediate to generate substituted quinolin-2(1*H*)-ones. The reduction conditions were mild and tolerant towards the functionality on both reacting partners. Though the Friedländer synthesis using 2-nitroaromatic ketones was previously performed in the presence of strong acids, the current results indicate that AcOH is an excellent solvent for both the current reaction from 2-nitroaromatic aldehydes and ketones as well as the classical variant from 2-aminobenzaldehyde.

## Data Availability

Not applicable.

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
