# Peer review of "Domino Nitro Reduction-Friedländer Heterocyclization for the Preparation of Quinolines"

_molecules, 2022, doi:10.3390/molecules27134123_

Round 1

Reviewer 1 Report

Khabena et al. describe synthesis of series of quinolines from nitro benzaldehyde by Friedlander reaction.

All the significant studies ( 1H, 13C spectra) were made.

in introduction to complete  brief of this organic reaction pls describe with "short" sentence also other modified friedlander reaction adding references  ( i.e. J. Org. Chem. 1995, 60, 22, 7369–7371, Bull. Korean Chem. Soc. 2005, Vol. 26, No. 2 323)

-Also if elemental analysis  is correct would be possible add some mass spectra of derivatives obtained?

-Could add in supporting information spectra IR of some compounds to demonstrate sentence ( line 151-154)

=================================

in intro, line 71-72: add (:)  after (below).

fig1: mefloquine , not mefloquin

fig1: if show stereochemistry for camptothecin, so add also stereochemistry for mefloquine

-line82 : pentanedione (2)?, in scheme 1 product 2 is not 2, 4-pentanedione

also in scheme 3 : correct pentanedione (2)

-pls update references for malaria ( 17-23) and tubercolosis (25) with recent papers/review

hence this reviewer accept this manuscript after major revision for publication in Molecules 

Author Response

Thank you for your comments and suggestions.

Though I have not done a thorough search for modified Friedländer methods, I have added a short sentence and added the references you mentioned.

I have not added HRMS data since they are very expensive and only indicate that the compound is present. I felt that elemental analyses, which also gives an indication of purity, were more useful.

All of the IR data are listed in the SI.  All of the relevant peaks are reported.

Since one should not end a sentence with a colon, I have retained the punctuation (a period) at the end of the sentence (line 71-72).

I have changed mefloquin to mefloquine as indicated in Fig. 1.  As to the stereochemistry of this drug, it is used as a racemic mixture of the R,S and S,R enantiomers.  I have drawn one of the enantiomers, but have indicated that the commercial drug is racemic.

The 2,4-pentanedione (acetylacetone) is correct.  I have drawn out the second methyl ketone as a substituent on the acetone portion of the molecule.  In Scheme 1, 2,4-pentanedione (2) is a reactant, not a product.  In Scheme 3, 2,4-pentanedione is shown only by number.  This proceeds to its enol form (pictured) during the reaction.

All of the references we reported regarding malaria and tuberculosis have been published during the past 10-12 years.  Several of these are review articles that show historically how these compounds were developed and modified.  This seemed more efficient that referencing numerous individual studies.

Reviewer 2 Report

This manuscript presents a very useful modification of the Friedlander synthesis protocol, and should find immediate use by medicinal chemists. Overall, the manuscript is well written. I only found one typo (the first compound of Scheme 2 has an extra minus charge on the carbonyl). The intro gave a good background of previous methods and justified the need for a reduction-based approach. I also appreciated the detailed mechanistic discussion and analysis. Great substrate scope and results. This draft is essentially publishable as is after fixing the minor typo in Scheme 2.

Author Response

Thank you for your kind comments.  I have corrected the typo in Scheme 2.

Reviewer 3 Report

Bunce and Fobi reported an experimental study on reductive cyclization protocol for the synthesis of quinoline derivatives. They showed that quinolines can be synthesized from 2-nitrobenzaldehydes and active methylene compounds under the action of Fe/AcOH system. However, such modification of Friedländer heterocyclization was already presented by many authors. The same protocol (Fe powder, acetic acid) was used into the following articles: RSC Adv. 2014, 37806-37811; Dyes and Pigments 2020, 108710; Org. Lett. 2017, 2841–2844 (see SI). Therefore, the originality and novelty of the presented manuscript was rated as very low. In view of the high quality of Molecules journal, I do not recommend this manuscript for publication.

Some comments that can be useful when resubmitting a manuscript to another journal:

1. The numeration of the synthesized compounds should be added.

2. The chemical structures should be added on NMR spectra.

Author Response

Thank you for your comments and suggestions.

I note your comments regarding previous studies of this reaction.  Of the three papers cited, only one is a targeted study that should have been found.  The Dyes and Pigments manuscript included only one reaction to prepare a quinoline-fused fluorescent dye, and otherwise, the paper has nothing that overlaps with our manuscript.  The Org. Lett. manuscript had all of its cases reported in the SI with no mention of these in the manuscript itself.  The RSC Adv. manuscript reports a number of reactions that overlap with the current study;  however, the number is not large.  The packaging of this manuscript as a "tandem C–C and C–N bond formation" did not identify it as a relevant study using any of the available search engines.  The manuscript focuses only on reactions that use 2-nitrobenzaldehydes with a limited number of active methylene compounds.  Our manuscript includes a greater breadth of functionalization of these activated reaction partners.  Additionally, the RSC Adv. manuscript also lacks examples involving 2-nitroaromatic ketones.  Thus, more heavily substituted targets were not prepared and the competing reaction to form quinolin-2(1H)-ones was not be detected.  Additionally, little was said about the mechanism of any of the reactions.  Thus, I believe our manuscript is a reasonable contribution to the literature in this area.  If all scientific knowledge was based on singular studies, very little would be known about anything.  I have, of course, added citations to the two publications that report the earlier work in this area; our original literature search did not identify either as having any relation to our planned study.

Structures have been placed on each set of spectra to allow readers to find them more readily.

Round 2

Reviewer 3 Report

The manuscript can be published